# RoBoCoP: A Comprehensive ROmance BOrrowing COgnate Package and Benchmark for Multilingual Cognate Identification

**Liviu P. Dinu**[♠,♡] **Ana Sabina Uban**[♠,♡] **Alina Maria Cristea**[♡]
**Anca Dinu**[♣,♡] **Bogdan Iordache**[♡] **Simona Georgescu**[♣,♡] **Laurențiu Zoicaș**[♣,♡]

University of Bucharest, ♠ Faculty of Mathematics and Computer Science,
♣Faculty of Foreign Languages and Literatures, ♡HLT Research Center

{ldinu, auban, alina.cristea}@fmi.unibuc.ro, iordache.bogdan1998@gmail.com,
{anca.dinu, simona.georgescu, laurentiu.zoicas}@lls.unibuc.ro

## Abstract

The identification of cognates is a fundamental process in historical linguistics, on which any further research is based. Even though there are several cognate databases for Romance languages, they are rather scattered, incomplete, noisy, contain unreliable information, or have uncertain availability. In this paper we introduce a comprehensive database of Romance cognates and borrowings based on the etymological information provided by the dictionaries (the largest known database of this kind, in our best knowledge). We extract pairs of cognates between any two Romance languages by parsing electronic dictionaries of Romanian, Italian, Spanish, Portuguese and French. Based on this resource, we propose a strong benchmark for the automatic detection of cognates, by applying machine learning and deep learning based methods on any two pairs of Romance languages. We find that automatic identification of cognates is possible with accuracy averaging around 94% for the more difficult task formulations.

## 1 Introduction and Related Work

Cognates detection and discrimination, as both the foundation of historical linguistics (Campbell, 1998; Mallory and Adams, 2006) and the starting point in historical investigation (Mailhammer, 2015), open windows on numerous areas of social sciences. The immediate implications of the accurate identification of cognate chains can be found in linguistic phylogeny (Atkinson et al., 2005; Alekseyenko et al., 2012; Dunn, 2015; Brown et al., 2008), allowing to trace back language relatedness (Ng et al., 2010) as well as linguistic contact (Epps, 2014), and offering important clues concerning the geographical and chronological dimension of ancient communities (Heggarty, 2015; Mallory and Adams, 2006). The cognate chains are the foundation of the "comparative grammar-reconstruction" method (Chambon, 2007; Buchi and Schweickard,

2014), and the etymological data thus obtained can be used as a source on human prehistory, corroborating the archaeological inventory (Heggarty, 2015), and providing the basis for 'linguistic paleontology' or 'socio-cultural reconstruction' (Epps, 2014). An extensive perspective on cognate chains can serve as a basis in the detection of meaning divergence, especially when searching for common patterns that govern the cognitive mechanisms activated in semantic change (Dworkin, 2006). The lexicon still offers significant clues for building a 'universal' cognitive network derived from meaning shifts, easily observable in cognate sets, which would be essential in developing a comprehensive theory on cognition and neuropsychology (Glessgen, 2011). At the same time, an integrated view on the cognate pairs between any two related languages would allow taking steps forward in the study of language acquisition (Huckin and Coady, 1999), as well as in the difficult task of eliminating false friends in automatic translation (Uban and Dinu, 2020).

Training the machine towards an accurate detection of cognates becomes a necessity in today's large amount of linguistic data that still hasn't been processed from a historical point of view (List et al., 2017). Since the foundation of the "comparative grammar-reconstruction" method over two centuries ago, linguistic phylogeny is still mainly investigated by means of manual comparison of cognate sets, which implies the extraction of systematic phonetic correspondences between words in language networks, and eventually allows the reconstruction of the protolanguage. Although successfully used by classical linguists, this method is highly counter-economic, which has led to a sustained search for computational methods able to assist the process. The increasing interest in automatic methods for cognate detection calls for a directly proportional need for reliable databases of positive examples, consisting of lists of cognate

sets as long as possible. These lists are not easily attainable, even if we deal with well-known languages, with well-developed electronic resources, like the Romance languages[1].

In order to obtain longer lists of cognate sets, the definition of cognates has broadened its limits, including: a) words sharing a similar form and meaning, regardless of their etymology (Frunza et al., 2005; Frunza and Inkpen, 2006, 2008), also referred to as 'true friends' (Fourrier and Sagot, 2022; Fourrier, 2022) (e.g. Eng. *famine* and Fr. *famine*, although the first one is borrowed from the second); b) words etymologically related, regardless of the type of relation (Hämäläinen and Rueter, 2019) (e.g. Eng. *family*, borrowed from Middle French *famile*, in its turn borrowed from Lat. *familia*, and Rom. *femeie* 'woman', inherited from Lat. *familia*); c) words that are similar in their orthographic or phonetic form and are possible translations of each other (Kondrak et al., 2003) (e.g. Eng. *sprint* and Japanese *supurinto*; see above, (Frunza and Inkpen, 2006)).

Besides these interpretations, there were also attempts to define cognates by establishing unnecessary limits: words that share a common origin and have the same English translation (Wu and Yarowsky, 2018). Following this acceptation, Ro *pleca* 'leave' and Es *llegar* 'arrive' are not to be identified as cognates, although they are both inherited from Lat. *plicare* 'to fold'; such narrowing disregards the possible benefits of a comparative perspective on cognates in the analysis of semantic divergence, one of the most understudied and promising fields in historical linguistics.

The simplest definitions of cognates, as "words sharing a common proto-word" (Meloni et al., 2021) or "words that share a common etymological origin" (Fourrier and Sagot, 2022) are likely to be the most effective in computational linguistics, although certain improvements can be made. In this paper, we use the following definition: two words are cognates if and only if the intersection of the sets of their etymons is not void.

The urge for a wide (if not exhaustive) database of the Romance lexicon derives both from internal and external needs. On the one hand, the high number of parallel chains of cognates would allow the Romance linguists to revisit the issue of sound laws, which, although apparently well-known, still raises questions about features whose correspondence is not regular and, therefore, is not easily explainable (e.g. Lat *signum / signa*, Ro *semn*, Es *seña*, *vs* Lat *sifflare*, Ro *sufla*, Es **ch**illar, where the latter form is considered to be irregular, although this phonetic evolution is not limited to an isolated number of instances). Moreover, in certain cases where there aren't enough data, linguists still argue whether a phonetic change represents the rule or, on the contrary, it's the apparent exceptions that make the rule (e.g. Lat ***fl**amma* > Es *llama*, but Lat ***fl**ore-* > Es ***fl**or*: at the moment, both theories are based on an even number of examples). Additionally, a complete diagram of the possible phonetic shifts would authorize etymologists to bring back into discussion long-standing etymological cruxes.

As for the external needs, we postulate that the algorithms identified by training the machine on one of the best studied language families could be further successfully applied to other languages which are less known or have scarce resources.

In terms of automatic approaches for cognate detection, the last decades bring a plethora of such methods (Rama et al., 2018; Jäger et al., 2017; Ciobanu and Dinu, 2014b; Fourrier and Sagot, 2022; Frunza and Inkpen, 2008; Mitkov et al., 2007). Most methods proposed in previous studies include linguistic features and different orthographic and phonetic alignment methods in combination with shallow supervised machine learning models (such as SVMs) or clustering methods (Bergsma and Kondrak, 2007; Inkpen et al., 2005; List, 2012; Koehn and Knight, 2000; Mulloni and Pekar, 2006; Navlea and Todirascu, 2011; Ciobanu and Dinu, 2014b; Simard et al., 1992; Jäger et al., 2017; St Arnaud et al., 2017). A few studies employ deep learning for cognate detection or related tasks. Rama (2016) use siamese convolutional neural networks (CNNs) with character and phonetical features complemented with additional linguistic features in order to detect cognates in languages across three language families, with the majority of examples belonging to Austronesian languages, with up to 85% accuracy. Miller et al. (2020) use language models including a recurrent neural network architecture for lexical borrowing detection. Transformers were used in (Celano, 2022) for predicting cognate reflexes. To the best of our knowledge, no previous studies have used transformer architectures specifically for cognate detection. Previous results on Romance cognate

---

[1] Among the more than 7000 languages of the world, barely 200 are provided with electronic resources (cf. (Bird, 2020))

detection in particular are reported in (Ciobanu and Dinu, 2014b), in which cognates are automatically distinguished from non-cognate translation pairs, based on a smaller dataset of cognate pairs in the Romance languages, with accuracies reaching 87% using an SVM with alignment features.

Starting with these remarks, our main contributions are:

1. We introduce a comprehensive database of Romance cognate pairs (pairs of cognates between any two Romance languages), by parsing the electronic dictionaries with etymological information of Romanian, Italian, Spanish, Portuguese and French (the database will be available for research purposes upon request).

2. We propose a strong benchmark for the automatic detection of cognates, by applying a set of machine learning models (using various feature sets and architectures) on any two pairs of Romance languages.

The rest of the paper is organized as follows: in Sections 2 and 3 we present the database which we have created and offer details about the processing steps involved, in Section 4 we introduce our approach for the automatic detection of cognates. Methodological details are discussed in Section 4.1, and an extensive error and results analyses is presented in Section 4.2. The last section is dedicated to final remarks.

## 2  Dataset

Even though there are several cognate databases for Romance languages, they are incomplete (as the inventory of Romance lexemes based on the Swadesh list (Saenko and Starostin, 2015), cf. (Dockum and Bowern, 2019)), noisy (because of the lack of expert proofing, these being usually obtained with the help of volunteers, like Wikipedia (Meloni et al., 2021), built with automated translation methods (Dinu and Ciobanu, 2014; Wu and Yarowsky, 2022), or are of uncertain availability (cf. (List et al., 2022)).To overcome as much as possible these weaknesses, we have decided to build from scratch a fully available database of Romance cognates, for the main five Romance languages (Italian - It, Spanish - Es, French - Fr, Portuguese - Pt and Romanian - Ro), starting with the available machine-readable reference dictionaries[2], which

contain etymological information. The process was semi-automated, guided and verified by human experts, to ensure the quality and coverage of the data.

Our strategy was to parse one by one all the dictionaries, to extract for each language exhaustive information related to every word and its relevant etymology features (namely its etymon(s), the source language(s), the part(s) of speech), and then to aggregate all this information in order to build a cognate database for all five Romance languages (from now on called RoBoCoP - Romance Borrowings Cognates Package) (see section 2.2). Since each of the five dictionaries had its own editorial choice of presenting the information, the preprocessing, the parsing and the postprocessing strategies had to be customized for each language, which implied a lot of expert and computational effort. The process was very specific to each dictionary and included a cyclical process similar to methodologies used in web scraping - running scripts implementing rule-based algorithms (such as regular expressions) to separate noise from the data for each dictionary and manual evaluation of each output with the assistance of linguists in our team, followed by potential refinement of the code to manage all exceptions. Due to the lack of space, we cannot present in detail all the challenges of building the RoBoCoP database, but we only discuss some of the most common difficulties. Addressing them all was a repetitive feedback process involving linguists and computer scientists.

### 2.1  Data Cleaning and Preprocessing

The preprocessing included cleaning and normalization. We always preserved in our database all accents, diacritics and any other characters that are part of the orthography of the words and etymons. We only normalized additional characters which are occasionally used to indicate pronunciation for the etymons in the source dictionaries (for example, in Romanian, accents are never part of the spelling of the word, but they can occur in the dictionary in order to indicate the stressed syllable). We additionally preserved the etymons exactly as encountered in the dictionary (pre-normalized) in

---

[2]Italian: *Il dizionario della lingua italiana De Mauro*, dizionario.internazionale.it.
Spanish: *Diccionario de la lengua española* published by *Real Academia Española*, lema.rae.es/drae.

Portuguese: *Dicionário infopédia da Língua Portuguesa*, published by Porto Editora, www.infopedia.pt/lingua-portuguesa.
French: *Trésor de la Langue Française informatisé* published by *Centre National de Ressources Textuelles et Lexicales*, www.cnrtl.fr.
Romanian: *Dicționarul Explicativ al Limbii Române* published by *Academia Română*, dexonline.ro.

| Word | Source language | Raw Etymon | Etymon |
|---|---|---|---|
| aeroplano | french | aéroplane | aéroplane |
| aerosol | french | aérosol | aérosol |
| afabilidad | latin | affabilĭtas | affabilitas |
| afable | latin | affabĭlis | affabilis |

Table 1: Excerpt from Spanish etymology dictionary in the RoBoCoP package.

a separate column in the database, in case they can be useful for future applications. Moreover, we used the full form of the words including pronunciation indications as the input for generating phonetic transcriptions. We only applied accent and diacritics removal as part of the classification experiments when extracting graphical features.

Table 1 illustrates a selection of a few example rows from our database from the Spanish etymology dictionary.

We also manually identified the meaning of the abbreviations used throughout the dictionaries, where such a list was not provided in the dictionary. The parsing process was by far the most difficult for French. The biggest challenge was the analytical presentation of the etymological information, organized as a summary of the history of the word, which resulted in a very complex parsing process.

The difficulties encountered by the machine in the cognate identification can be classified in two sub-types: 1) cognates whose etymon was registered under different paradigmatic forms, e.g. for nouns, nominative *rex*, *vs* accusative *regem*, leading to missing cognate pairs, such as Es *rey* (< Lat. *rex*) - Fr *roi* (< Lat. *regem*); 2) cognates whose etymologies do not correspond from the point of view of the diachronic or diastratic specifications, e.g. Es *local* (< Lat *localis*) - Fr *local* ("emprunté au lat. de basse époque *localis*", "borrowed from Late Latin *localis*"): the machine did not match the abreviations "Lat" with "Late Lat".

To overcome the first problem, we added an additional preprocessing step of lemmatizing all the Latin etymons using the CLTK library[3] (Johnson et al., 2021), thus recovering 13,227 cognate pairs in total. We only applied etymon lemmatization as part of the cognate matching algorithm, and kept in our database the original etymon as found in the dictionary, in case the information can further serve for other applications. The second problem led to the necessity of extracting and sorting the

source languages. Each dictionary used its own way of abbreviating a source language, e.g. *tc.*, *tur.*, *turc.*, *turk.* all refer to the Turkish language. We manually normalized language abbreviations across dictionaries, as well as collapsed some of the language varieties with the help of linguists, resulting in a fixed set of source languages that are necessary and sufficient for identifying Romance cognates in a linguistically justified manner (e.g. the Languedocian and the Limousin were collapsed as Occitan, being both dialects of this language). We also leveraged the diachronic and diastratic indications, compiling in the end a list of 259 total identified source languages. Once we extracted the etymologies for each word and language, we moved to the construction of the cognate database, standardizing and structuring the extracted information, so it can be further accessed easily for a wide range of experiments. We describe the construction process in more detail in the next subsection.

## 2.2 The Construction of the RoBoCoP Database

For each of the five Romance languages (It, Es, Fr, Pt, Ro), the database contains lists of words, with their etymologies. Starting with these data, we obtained new lists of cognate pairs between any two Romance languages of the five, by the following procedure. For any triplet <*u, e, L_1*> in language $L_1$, if we find a triplet <*v, e, L_2*> in $L_2$ (having the same etymon *e*), add the triplet <*u, v, e*> to the list of cognate pairs of the language pair ($L_1$, $L_2$). We define two words in a pair of languages as being cognates if and only if the intersection of the sets of their etymons is not void. This definition is the most general and in line with other previous definitions used (Ciobanu and Dinu, 2014b, 2019; Fourrier, 2022). Because RoBoCoP covers both words with Latin origin and borrowings of different origins, it is easy to retrieve from its content more specific definitions. One such definition minds only the most distant ancestor word, the Latin one. To comply with this definition, one needs only to add to RoBoCoP the constraint that the only source language should be Latin and thus one removes the more recent borrowings from other languages.

Another particular definition states that two words are cognates if they have a common ancestor, regardless of the level. For example, if two words, *u* from language *A* and *v* from language *B* have two different etymons, $e_1$ and $e_2$, respectively,

---

[3]https://docs.cltk.org/en/latest/_module%s/cltk/lemmatize/lat.html

| Ro | It | Es | Pt | Fr |
|---|---|---|---|---|
| 45465 | 24257 | 16458 | 28180 | 19822 |
| fr:35511 | lat:18437 | lat:11936 | lat:17446 | lat:12804 |
| lat:9312 | fr:1981 | fr:1366 | gr:2818 | en:1086 |
| it:3358 | gr:1649 | es:712 | fr:2369 | it:912 |

Table 2: Number of words in dictionaries for each language (upper row), and most frequent source language (lower row) across words in a dictionary.

and $e_1$ and $e_2$ have the same etymon $e$, then, by transitivity (Batsuren et al., 2022), the words $u$ and $v$ are cognates. For instance, this happens often for the language pair (Ro, Pt). This definition of cognates can also be recuperated programmatically from the RoBoCoP database. Both these particular cognate definitions were difficult to account for using other cognate resources, rendering comparison of the computational methods of cognates identification cumbersome or even impossible. As previously stated, for the purposes of this study, we use the definition of cognates where two words are cognates if and only if they share a common etymon (at the first level). Nevertheless, our database supports any of the three versions of cognate definition, becoming a valuable and attractive resource. Aside from cognate and borrowing identification, it allows for proto-words identification as well, which makes RoBoCoP not only a multilingual, but also a powerful multitasking resource. Moreover, since the source of this database are dictionaries of Romance languages, that implicitly include all the words currently in use, it has a wide coverage of cognate pairs of any two Romance languages from the five included, thus maximizing recall. Another advantage of the database is that it minimizes the noise and is reliable, because it was obtained in a computer - assisted manner and manually checked, as opposed to other Romance cognates resources created from Wiktionary or from automated translations.

## 3 Quantitative Aspects of the Database

We list here some quantitative aspects of the database. The database comprises a total of 125,598 words across all languages and 90,853 cognate pairs. Table 2 shows the total number of words per language and the top three source languages for borrowings, for each language. The number of cognate pairs identified for any language pair is depicted in Table 3.

Regarding the accuracy of the extraction and

|  |  | It | Es | Pt | Fr |
|---|---|---|---|---|---|
| Ro | all | 6,683 | 9,056 | 8,211 | 8,120 |
|  | lat | 4,999 | 7,588 | 5,855 | 7,360 |
| It | all |  | 8,627 | 13,343 | 7,361 |
|  | latin |  | 7,863 | 12,198 | 7,105 |
| Es | all |  |  | 10,731 | 10,543 |
|  | lat |  |  | 9,533 | 10,220 |
| Pt | all |  |  |  | 8,179 |
|  | lat |  |  |  | 7,783 |

Table 3: Number of cognate pairs for each language pair: total number and pairs of Latin etymology only.

|  | Ro | It | Es | Pt |
|---|---|---|---|---|
| It | 98% |  |  |  |
| Es | 98% | 99% |  |  |
| Pt | 99% | 99% | 97% |  |
| Fr | 98% | 98% | 98% | 98% |

Table 4: Estimated accuracy (based on 100 randomly sampled cognate pairs for each language pair) for our cognate extraction method used for building our database based on etymology dictionaries.

cleaning algorithm, we have computed accuracy scores based on random samples of 100 entries in each language's etymology dictionary in our database, as well as for each list of cognate pairs corresponding to all language pairs in our database.

We find the following accuracies for extracting etymologies for each language (the average accuracy is 98.6%): Spanish: 100%, Romanian: 98%, Portuguese 97%, Italian 100%, French 98%. For cognates extraction we find the following accuracies for each language pair as seen in Table 4 (with an average of 98.2%).

As reflected by the quantitative aspects provided in this section, we created both an effective resource for Romance cognate pairs, and a comprehensive map of borrowings and etymologies for the Romance languages. RoBoCoP is, to our best knowledge, one of the most high-coverage, reliable and complex databases of Romance cognates.

### 3.1 Comparison with other Romance Cognates Databases

By comparison with other Romance cognates resources, our database turns out to be more inclusive and well-grounded, as well as the most comprehensive, to the best of our knowledge. The database in (Bouchard-Côté et al., 2007) only comprises 3 Romance languages, Italian, Spanish and Portuguese and contains a much smaller number of cognate pairs. By contrast, RoBoCoP also includes French and Romanian, and defines cognates most generally, with the possibility of recuperating any

of the more restrictive definitions. The resource in (Ciobanu and Dinu, 2014b, 2019) contains only cognate pairs between Romanian and all other main Romance languages. Also, the method used for identifying the cognates employed an intermediary step of Google Translation. Another database (Meloni et al., 2021) that starts from the one proposed in (Ciobanu and Dinu, 2014b) by adding only those cognate pairs from Wiktionary that have a common Latin ancestor. Compared to this data set, RoBo-CoP has much more cognate pairs. The archive in (He et al., 2022) uses the work of (Meloni et al., 2021), but removes inexplicably the Romanian language. Finally, the database in (List et al., 2022) covers more languages, but with much fewer cognate pairs than ours, for any language pair.

## 4   Automatic Cognate Detection Experiments

During the last decades, several computational approaches to the automatic detection of cognate pairs reported fairly good results. The main problem when it comes to evaluating them is that, almost always, the results are not directly comparable across studies. This is not only due to the application of different methodologies, but, most of the time, due to the use of different databases. We address this issue by proposing a comprehensive, reliable database of Romance cognates.

In the following, we will present a series of experiments and results to further help the evaluation process of automatic cognate pairs detection, providing a benchmark for future approaches.

### 4.1   Methodology

We frame the problem as a binary classification problem, where cognate pairs are positive examples. Because the ultimate purpose of all the experiments is to decide if a pair of words are cognates or not, one needs for training both positive data (pairs of cognates provided by RoBoCoP) and negative data (pairs of non-cognate words). It is remarkable that, to our best coverage of the literature, while positive data was generally well documented, negative data lack explanations, with a few exceptions (Ciobanu and Dinu, 2014b). The choice of negative examples is essential in informing the interpretation of automatic detection results. For instance, it is easy to decide that two obviously different words in two languages such as Romanian *apă* ('water') and Spanish *cerveza* ('beer') are not cognates, but

not so easy for more similar words such as Italian *rumare* ('rumble') and Romanian *rumen* ('ruddy').

**Negative examples.**   To address this issue, we propose two methods of negative example generation which we consider in all experiments.

*Random negative sampling.* In the simpler setting, we generate a negative cognate pair selection that contains pairs of words randomly extracted from non-cognate pairs.

*Levenshtein-based negative sampling.* We include a second method where we select as negative examples graphically similar word pairs which do not have common etymology, by conditioning the words in the pair to have a Levenshtein distance (Levenshtein, 1965) smaller than the average Levenshtein distance across cognate sets for that language pair. The Appendix illustrates in more detail the distribution of Levenshtein distances across cognate pairs. Thus, the average Levenshtein distances for negative pairs are smaller than those of positive pairs, ensuring that distinguishing between them is not trivial based on their form. In both settings, we sample words forming negative pairs from word lists in our dictionaries across the entire vocabulary (including inherited and borrowed words, as well as words formed internally). We also include in our database our selection of negative examples in order to facilitate reproductibility.

**Experimental settings.**   For all language pairs, we generate datasets balanced in positive and negative examples. We use a $80\% : 20\%$ split to generate train and test sets, which are initially shuffled. For validation we use 3-fold cross validation on the training data for all experiments, unless explicitly stated otherwise. We perform a separate set of experiments where we limit positive examples to words with Latin etymology. Negative examples are sampled from non-cognates in a similar way, maintaining the balance between classes.

**Features.**   All the experiments are performed using both the graphic form of the words and the phonetic one. To obtain the latter, we employed the eSpeak library[4], a resource used also by other similar studies (Meloni et al., 2021). For some of our experiments we include feature extraction consisting of computing alignments on the word pairs, emulating methods used by historical linguists. Ciobanu and Dinu (2019) showed that extracting features from the alignment returned by the Needleman-Wunsch

---

[4]https://github.com/espeak-ng/espeak-ng

|     |     | Ro | | | It | | | Es | | | Pt | | | Fr | | |
| --- | --- | --- | --- | --- | --- | --- | --- | --- | --- | --- | --- | --- | --- | --- | --- | --- |
|     |     | Gr | Ph | En | Gr | Ph | En | Gr | Ph | En | Gr | Ph | En | Gr | Ph | En |
| Ro | Lat | - | - | - | 93.8 | 93.9 | **94.8** | 90.6 | 90.7 | **92.1** | 91.8 | 91.1 | **92.7** | 90.5 | 89.6 | **91.9** |
|     | All | - | - | - | 92.4 | 93.6 | 94.6 | 90.4 | 89.8 | 91.2 | 91.6 | 90.5 | 92.5 | 90.1 | 89.2 | 91.2 |
| It | Lat | 98.5 | 98.0 | 98.6 | - | - | - | 93.9 | 94.0 | 95.1 | 94.2 | 94.1 | **95.6** | 93.4 | 93.2 | **95.0** |
|     | All | **98.8** | 97.9 | 98.7 | - | - | - | 93.9 | 94.3 | **95.2** | 94.3 | 93.7 | 95.2 | 93.0 | 92.2 | 94.2 |
| Es | Lat | 98.5 | 97.8 | **98.7** | 98.5 | 98.3 | **99.0** | - | - | - | 91.3 | 91.2 | 92.2 | 93.3 | 93.0 | 95.0 |
|     | All | 98.3 | 97.5 | 98.5 | 98.8 | 98.3 | 99.0 | - | - | - | 91.7 | 90.7 | **93.1** | 93.5 | 93.0 | **95.1** |
| Pt | Lat | 98.7 | 97.8 | **98.8** | 99.0 | 98.7 | **99.2** | 98.2 | 97.9 | 98.7 | - | - | - | 93.8 | 92.9 | **95.2** |
|     | All | 97.9 | 97.9 | 98.5 | 98.8 | 98.4 | 98.9 | 98.1 | 97.6 | **98.9** | - | - | - | 92.7 | 93.7 | 95.1 |
| Fr | Lat | 98.5 | 98.0 | **98.9** | 98.9 | 98.4 | **99.1** | 98.5 | 98.3 | **99.1** | 98.1 | 97.6 | 98.7 | - | - | - |
|     | All | 98.5 | 97.9 | 98.7 | 98.7 | 98.2 | 98.8 | 98.3 | 98.0 | 98.9 | 98.2 | 98.2 | **98.8** | - | - | - |

Table 5: Classification accuracy on the test set using the ensemble model. For each language pair, the results for all cognate pairs as well as for pure cognate pairs only (Latin etymon) are displayed on two consecutive rows; the results using graphic-only (Gr), and phonetic-only (Ph) features, and the best ensemble (En) with combined features are shown on three consecutive columns. The results using the Levenshtein-based negative sampling are shown above the main diagonal, while the results using random negative sampling are shown below the main diagonal.

algorithm (Needleman and Wunsch, 1970) on the graphic representations of the words achieved good results when used for training machine learning models for cognates classification. We implement the same approach for extracting $n$-grams around mismatches from the alignment (caused by insertion, deletion, or substitution). Furthermore, for a given value of $n$, we consider all such $i$-grams with the length $i \leq n$. For example, given the French-Romanian pair (dieu, zeu), we obtain the alignment ($dieu$, $z-eu$), where $ marks the beginning and ending of the alignments and – represents a deletion/insertion. For $n = 2$, the extracted features would be d>z, i>-, $d>$z, di>z-, and ie>-e. These features are then vectorized using a binary bag of words. Unlike previous work, we also experimented with the alignment of phonetic representations.

**Ensemble Model.** Our first set of experiments involves training various machine learning algorithms on the alignment features computed for either the graphic or the phonetic representations. For the graphic representations we preprocess the words by removing accents. We experiment with various algorithms: Support Vector Machine, Naive Bayes, XGBoost classifier (Chen and Guestrin, 2016). These models are trained on either the graphic or the phonetic alignments using various hyper-parameters and their performance is assessed using cross validation. For each language pair, we select the best five performing algorithms and train a stacking ensemble classifier. In order to guarantee the presence of both graphic and phonetic features in the final ensembles we make sure to never select more than three models that were trained on

graphic, or phonetic features, respectively. We also evaluate ensembles trained using only graphic and only phonetic base models, respectively, to assess if any category of features outperforms the other, or if their combination is more favorable.

**Convolutional Neural Network.** For the deep learning experiments, we encode the graphic or the phonetic representation of the words as simple sequences of characters and train deep neural networks to extract features and provide predictions. These models are "alignment-agnostic", in order to see if they can outperform the handcrafted features. The first architecture we employ is a siamese convolutional architecture, combining two CNNs where each arm models one of the words in the pair. Each word is treated as a character sequence, where characters are encoded as learned dense vectors using an embedding layer. The character vocabulary is constructed separately for each language pair, accented characters are kept separately. The outputs of the two CNNs are then concatenated and passed to the final dense layer to produce a prediction.

**Transformer.** The second model employs a Transformer architecture (Vaswani et al., 2017). Either the graphic or the phonetic representation of a word is split into individual characters (without any normalization). For a given pair of words, the character sequences are concatenated with a special `[SEP]` token placed between them and a `[CLS]` token placed before the first sequence. The tokens are then positionally embedded and fed through a multi-layered Transformer encoder. The output generated by the last encoder layer for the `[CLS]` token is used for classification and passed through a feed-forward layer. For this method we hold the

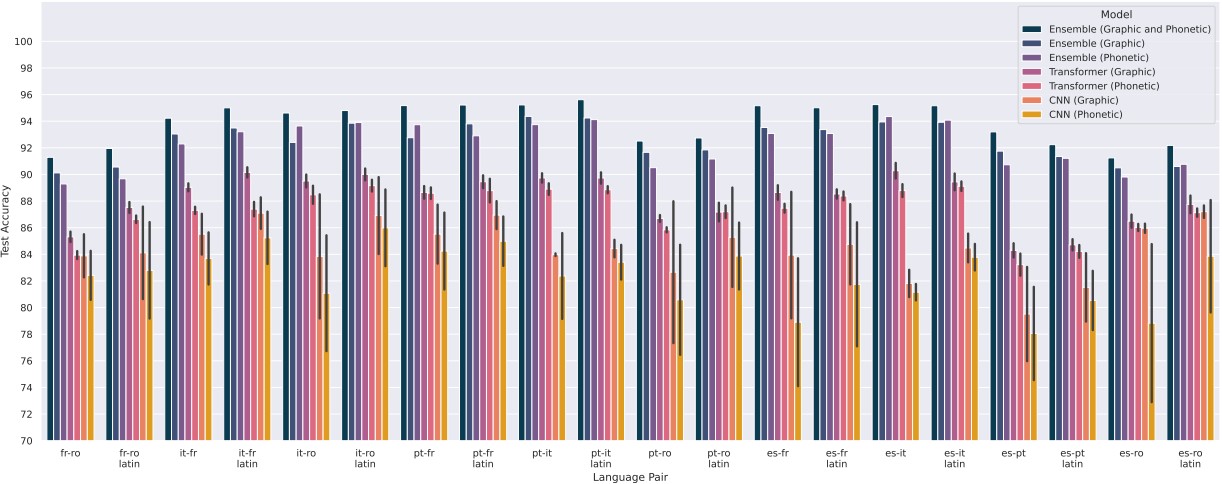

Figure 1: Test accuracy scores for all models and features, measured on full dataset and on pure cognates (Latin etymology), using close Levenshtein distance for negative example selection. Deep learning results computed as average across 5 runs.

last eighth of the training samples for validation. The validation dataset is also used for selecting the best model at each training epoch. For both deep learning approaches, we train independent models for each language pair, separately on graphical and phonetic representations of the words. Extensive details on the hyper-parameters and the training techniques employed are presented in the Appendix.

**Metrics.** Since datasets are balanced, we use accuracy as our main metric for model selection as well as for interpreting results, and additionally measure and report precision and recall. Detailed results are included in the Appendix.

### 4.2 Results and Error Analysis

The best results are obtained using the ensemble models with alignment features across all experimental settings, while the transformer-based model generally comes second (Figure 1). Table 5 shows results for the best performing model (the ensemble model) in all settings. This comparison shows that although no alignment method is superior in every scenario, the combination of both graphic and phonetic features in the ensemble model surpasses the ensembles that were limited to only one kind of features. For any pair of two languages, the cognate detection in random lexical pairs with our best model was accurate in a proportion of 98%-99%, with the lowest accuracy 98.5 for Es-Ro, and the highest of 99.07 for Es-It. When using Levenshtein-based negative sampling, we obtain a minimum of

91.24 for Es-Ro, and a maximum of 95.24 for It-Es. This confirms that identifying cognate pairs out of a selection containing clearly different words can be almost trivial for machine learning models. In Figure 1 we separately illustrate results obtained using all models and features and the second, more strict setting for generating negative pairs (Levenshtein-based sampling), which leads to more interesting variations.

We notice that, in both situations, Romanian provides the lowest values when compared with any other Romance language, while at the opposite end of the spectrum we find Italian. The interpretation of these results leads to a global perspective on the degree of similarity between languages, which was theoretically discussed in (Dinu and Dinu, 2005; Ciobanu and Dinu, 2014a), and is now able to favor a deeper insight into the phonetic structure of Romance languages measured in relation to Latin and in comparison with each other. While a usual suspect for poor results for low-resource languages such as Romanian is a scarcity of training data, in our case this can not be the case, since our dataset covers all cognates in the vocabulary exhaustively, so detection performance can not be improved simply by augmenting the training data. When taking into account only the cognates originating in Latin, we obtained, in most cases, even better results than for the full set: the lowest accuracy for randomly selected lexical pairs with the ensemble model was 98.65 for It-Ro, while the highest was 99.2 for Pt-It; when using Levenshtein-based negative sampling, the lowest accuracy was 91.9 for Fr-Ro, and the

| | Ro | It | Es | Pt | Fr |
|---|---|---|---|---|---|
| Ro | | e$>-$, re>–, ar>a- | o$>-$, r$>-$, ar>a- | r$>-$, ar>a-, o$>-$ | e$>-$, r$>-$, er>a- |
| It | e$>-$, re>–, ar>a- | | -$>e$, r->re, n->ne | -$>e$, r->re, –>ne | e$>-$, re>r-, ne>n- |
| Es | o$>-$, ar>a-, r$>-$ | -$>e$, r->re, n->ne | | e$>-$, on>o-, n$>-$ | ci>ti, se>–, ar>er |
| Pt | r$>-$, ar>a-, ca>ti | -$>e$, r->re, –>ne | e$>-$, on>o-, n$>-$ | | ao>io, o->on, ca>ti |
| Fr | e$>-$, r$>-$, er>a- | e$>-$, re>r-, ne>n- | ci>ti, se>–, ar>er | ao>io, o->on, ca>ti | |

Table 6: Top 3 informative graphic alignment bigrams according to $\chi^2$ feature selection, based on the full training dataset (above the main diagonal), and the training dataset containing only cognates of Latin origin (below the main diagonal). Bigrams are separated by commas, $>$ marks where the bigram for the first word in the pair ends and where the bigram for the second word begins, $-$ marks an insertion/deletion computed by the alignment algorithm.

highest 95.6 for Pt-It. This increasing accuracy is supported by a higher degree of regularity in the phonetic evolution from Latin to the Romance languages, which also leads to a better correspondence between any two Romance languages. It is thus obvious that the machine was able to better learn and recognize the phonetic correspondences between words inherited or borrowed from Latin, which were not applicable to borrowings from other Indo-European languages (such as English) or non-Indo-European idioms (such as Turkish or Arabic).

There are some cases where the identification of a pair of words as cognates was reported as an error, despite their obvious genetic relation. For example, Es *cognitivo* and Ro *cognitiv* are not registered as cognates in our database because they appear in dictionaries with different etymologies: the Spanish word is considered as an internal creation (a derivative from Es *cognición*), while the Romanian lexeme is a borrowing from Fr *cognitif*. The automatic selection of such word pairs as cognates calls into question the supposed status of internal creation of lexemes such as Es *cognitivo*, given the limited possibilities of derivation with the suffix *-ivo* (in this case) in Spanish (cf. (Española, 2010)), as well as the significant influence of the French language on Spanish.

We additionally extract relevant features by selecting the top character bigrams according to their weights in the ensemble models. It is especially interesting to compare these features with the criteria generally used by historical linguists for identifying cognates. We find, for example, that of the top ranked orthographic cues, none occurs at the beginning of the word, while many of them occur at the end of the word. Table 6 contains a list of top relevant features.

## 5 Conclusions and Future Work

We introduced a comprehensive database (in graphic and phonetic form) and framework for the automatic analysis and detection of Romance cognates (the largest database of this kind, in our best knowledge, with 125,598 words across all languages and 90,583 cognate pairs).

Our framework is the result of collaboration between computer scientists and linguists and includes: a linguistically informed and computationally usable definition of cognate words, a methodology for extracting cognate pairs automatically in a robust way, a comprehensive dataset of word etymologies for Romance languages based on etymological information given by dictionaries, and a comprehensive database of cognate pairs, as well as benchmark results for automatic cognate detection, based on a series of machine learning experiments (using a variety of features and models: graphical and phonetical features, including prior feature engineering to obtain word alignment information, or alignment-agnostic, and several types of model architectures) for automatically detecting cognates.

For the most difficult task (cognate detection for Levenshtein-based negative sampling) we obtained an average accuracy around 94%.

In future work we intend to distinguish virtual cognates in the database and to complement experiments with discrimination between virtual and true cognates. Furthermore, we aim to investigate the discrimination between cognates and borrowings, also adding semantic features, as well as more phonetic features for each pair of Romance languages.

## Ethics Statement

There are no ethical issues that could result from the publication of our work. Our experiments comply with all license agreements of the data sources used. We make the contents of our package available for research purposes upon request.

## Limitations

There are a few limitations to our cognate extraction methodology that could be improved upon in

future work. First, distinguishing between oral and written Latin can further refine the types of etymological relations between words of Latin origin. In isolated cases, the normalization of Latin etymons has led to incorrect cognate pairs.

Furthermore, according to a wider definition of cognates, cognates extraction could be extended to include deeper relatedness levels. In our experiments reported in this paper, in the case of Latin etymologies, we consider all cognate pairs which have a common Latin ancestor (directly derived from Latin). An extended version of our cognate database can be obtained based on our published dictionaries with etymological information and list of source languages, using the following extended definition: For any pair of Romance languages, we consider all cognate pairs which have a common ancestor at any level. For example, the Ro-Pt pair $<u,v>$ was obtained from the pair $<x,y>$, because $x$ and $y$ have a second level common ancestor $z$, and, consequently, we consider $<u,v>$ a cognate pair.

Another clear limitation is that our database only covers the main Romance languages, and does not yet include other Romance varieties nor any other language families. In terms of cognate detection results, we expect that detecting cognate pairs across language families could be more challenging, and that our results are an overestimation of that (confirmed by the improved results on pairs of Latin origin).

In terms of cognate detection experiments, we acknowledge there are different architectures and feature sets to be used for cognate detection which could improve results in the case of deep learning models, and we invite other researchers to propose new methods and test them on our database. An explainability analysis of the deep models could also be interesting to understand to what extent they are capable of identifying "alignment" patterns based only on word forms. A classifier trained on all language pairs together could also reveal interesting commonalities across language pairs, as well as potentially obtain better results due to this.

## Acknowledgements

We warmly thank Bianca Guiţă for her help offered in building the database. We would like to thank the anonymous EMNLP reviewers for their insightful feedback.

Research supported by the Ministry of Research, Innovation and Digitization, CNCS/CCCDI UE-FISCDI, CoToHiLi project, number 108/2021, Romania.

Liviu P. Dinu and Ana Sabina Uban contributed equally to this work.

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

# A Appendix. Additional Results

## A.1 Classification Experiments. Experimental Settings

### A.1.1 Ensemble Models

In order to select the best base models to be put into the ensemble various machine learning models were trained using the *scikit-learn* Python library and 3-fold cross validated on the training dataset. The list of models and their parameters is the following (note that if not specified, all other hyperparameters are set to the defaults set in the 1.2.0 version of the library):

- Support Vector Machine (`SVC`): $kernel = rbf, C \in \{0.1, 1, 10\}$

- Linear Support Vector Machine (`LinearSVC`): $C \in \{0.1, 1, 10\}$

- Multinomial Naive Bayes

- XGBoost classifier (*xgboost* library version 1.7.3)

We evaluate each such model using either graphic or phonetic features, and using various values for the size of considered alignment $n$-grams ($n \in \{1, 2, 3\}$).

For each language pair and selection setting for the negative examples, we select the best performing five model configurations and train a `StackingClassifier` on the whole training set. This is our final ensemble model for this approach.

Training, cross-validation, and testing for all experiments had a combined running time of $\approx 2h$. Infrastructure use was CPU: "Ryzen 5 3600X", 3.8 GHz, 6 cores.

### A.1.2 Transformer Models

**Architecture.** The model was created based on the `TransformerEncoder` implementation from the *torch* Python library (version 1.13.1) and it has the following structure:

- embedding size: 200

- hidden state size: 200

- number of attention heads: 8

- number of layers: 4

- dropout layer after positional encoding, probability: 0.2

- trainable parameters: $\approx 980, 202$

- Computational budget: training, validation, and testing for all experiments had a combined running time of $\approx 5h$ for all 5 runs.

- Computational infrastructure:
    - CPU: "Ryzen 5 3600X", 3.8 GHz, 6 cores
    - GPU: Nvidia RTX 2060 Super, 1470 MHz, 8 Gb VRAM

**Training Details.** In order to prevent overfitting we evaluate the model after each epoch on the validation set and if after the last epoch there was no increase with respect to the best previously encountered loss we reduce the learning rate of the optimizer with a coefficient $\gamma$. After a number of consecutive epochs without improvement we stop the training (see "patience" parameter). The parameters for training are the following:

- number of epochs: 50

- batch size: 64

- loss function: cross entropy loss

- optimizer: Adam

- initial learning rate: 0.001

- $\gamma$: 0.6

- patience: 5 epochs

### A.1.3 Siamese CNN Model

The model was developed using the Tensorflow ($v.2.11.0$) and Keras ($v2.11.0$) frameworks for deep learning.

- Trainable parameters: $17, 473$

- Computational infrastructure: Haswell 2.4GHz Intel Core i7-4700HQ, 4 cores

- Computational budget: $\approx 30mins$/model training and evaluation on CPU; $\approx 40h$/all experiments per run.

**Optimal Hyperparameters**
Optimal hyperparameters were found by manual tuning for cross-validation accuracy optimization.

- embedding dimension: 16,

- filters: 160,

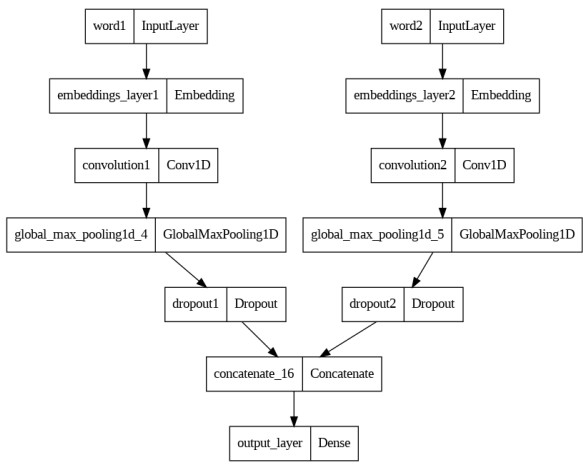

Figure 2: Architecture of siamese CNN model

- kernel size: 3,

- dropout: 0.1,

- $l_2$ dense: 0.00011,

- $l_2$ embeddings: 0.0000001,

- optimizer: adam,

- loss function: cross entropy loss,

- learning rate decay: 0.0001,

- learning rate: 0.005,

- early stopping patience: 5,

- max word length: 30,

- batch size: 160,

- epochs: 40

### A.1.4 Tools

- Phonetic transcription done using the *py-espeak-ng* Python wrapper (version 0.1.8) for eSpeak

## A.2 Database. Aditional results

## A.3 Classification Experiments Additional Results

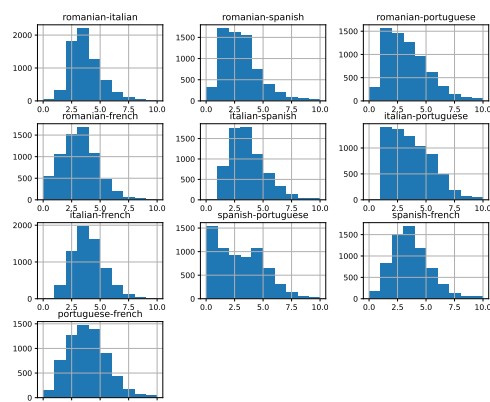

Figure 3: Distribution of Levenshtein distances for true pairs of cognates in our database.

| | | Ro | | | It | | | Es | | | Pt | | | Fr | | |
|---|---|---|---|---|---|---|---|---|---|---|---|---|---|---|---|---|
| | | Gr | Ph | En | Gr | Ph | En | Gr | Ph | En | Gr | Ph | En | Gr | Ph | En |
| Ro | Lat | | | | 93.4 | 94.5 | 95.4 | 91.0 | 91.0 | 92.6 | 90.1 | 91.4 | 92.1 | 90.7 | 90.1 | 92.3 |
| | All | | | | 92.0 | 93.2 | 94.6 | 88.3 | 88.5 | 90.5 | 90.9 | 90.8 | 92.6 | 88.3 | 89.7 | 91.6 |
| It | Lat | 98.2 | 98.4 | 98.5 | | | | 94.7 | 94.7 | 95.7 | 94.7 | 94.3 | 95.7 | 93.9 | 93.6 | 95.2 |
| | All | 98.9 | 98.4 | 99.1 | | | | 92.9 | 93.5 | 94.3 | 93.9 | 93.8 | 94.7 | 93.2 | 92.4 | 94.3 |
| Es | Lat | 98.7 | 98.4 | 99.0 | 98.7 | 98.6 | 99.1 | | | | 91.6 | 91.3 | 92.6 | 93.8 | 93.7 | 95.9 |
| | All | 98.3 | 98.1 | 98.7 | 98.8 | 98.2 | 99.0 | | | | 90.6 | 90.5 | 92.0 | 92.3 | 93.4 | 94.9 |
| Pt | Lat | 98.9 | 98.2 | 99.2 | 99.3 | 98.6 | 99.4 | 98.3 | 98.0 | 98.8 | | | | 94.4 | 94.1 | 95.8 |
| | All | 98.0 | 98.0 | 99.0 | 99.1 | 98.4 | 99.0 | 98.0 | 97.5 | 98.7 | | | | 92.7 | 94.8 | 95.8 |
| Fr | Lat | 98.6 | 98.0 | 98.9 | 99.0 | 98.3 | 99.2 | 98.5 | 98.4 | 99.2 | 98.4 | 97.9 | 98.9 | | | |
| | All | 98.9 | 98.2 | 99.1 | 98.8 | 98.5 | 98.8 | 98.0 | 98.3 | 99.1 | 98.4 | 98.5 | 99.3 | | | |

Table 7: Classification **precision** on the **test** set using the **ensemble** models, trained either exclusively using graphic classifiers (Gr), or phonetic classifiers (Ph) , or a combination of both (En). Scores above the main diagonal correspond to the Levenshtein distance-based negative samples selection, while scores below the main diagonal correspond to the random selection.

| | | Ro | | | It | | | Es | | | Pt | | | Fr | | |
|---|---|---|---|---|---|---|---|---|---|---|---|---|---|---|---|---|
| | | Gr | Ph | En | Gr | Ph | En | Gr | Ph | En | Gr | Ph | En | Gr | Ph | En |
| Ro | Lat | | | | 94.2 | 93.0 | 94.0 | 90.4 | 90.8 | 91.9 | 93.6 | 90.5 | 93.2 | 90.8 | 89.6 | 91.8 |
| | All | | | | 92.5 | 93.8 | 94.4 | 93.1 | 91.2 | 91.9 | 92.3 | 89.8 | 92.2 | 92.4 | 88.6 | 90.8 |
| It | Lat | 98.9 | 97.7 | 98.8 | | | | 93.3 | 93.6 | 94.8 | 93.9 | 94.1 | 95.6 | 93.3 | 92.9 | 94.9 |
| | All | 98.6 | 97.4 | 98.4 | | | | 94.8 | 95.1 | 96.0 | 94.7 | 93.5 | 95.6 | 92.7 | 92.0 | 94.0 |
| Es | Lat | 98.4 | 97.3 | 98.6 | 98.5 | 98.2 | 98.9 | | | | 91.3 | 91.3 | 92.0 | 93.3 | 92.8 | 94.3 |
| | All | 98.4 | 97.0 | 98.4 | 98.8 | 98.5 | 99.0 | | | | 92.8 | 90.5 | 94.2 | 94.6 | 92.3 | 95.2 |
| Pt | Lat | 98.6 | 97.3 | 98.5 | 98.8 | 98.9 | 99.2 | 98.2 | 97.9 | 98.6 | | | | 93.3 | 91.8 | 94.7 |
| | All | 97.9 | 97.7 | 98.1 | 98.6 | 98.5 | 98.9 | 98.3 | 97.8 | 99.0 | | | | 92.8 | 92.5 | 94.5 |
| Fr | Lat | 98.6 | 98.1 | 99.1 | 98.8 | 98.7 | 99.0 | 98.7 | 98.4 | 99.2 | 97.9 | 97.4 | 98.7 | | | |
| | All | 98.2 | 97.6 | 98.5 | 98.8 | 97.8 | 98.8 | 98.6 | 97.6 | 98.7 | 98.0 | 98.0 | 98.3 | | | |

Table 8: Classification **recall** on the **test** set using the **ensemble** models, trained either exclusively using graphic classifiers (Gr), or phonetic classifiers (Ph) , or a combination of both (En). Scores above the main diagonal correspond to the Levenshtein distance-based negative samples selection, while scores below the main diagonal correspond to the random selection.

| | | Ro | | It | | Es | | Pt | | Fr | |
|---|---|---|---|---|---|---|---|---|---|---|---|
| | | Gr | Ph | Gr | Ph | Gr | Ph | Gr | Ph | Gr | Ph |
| Ro | Lat | | | 90.0 | 89.2 | 87.8 | 87.1 | 87.2 | 87.2 | 87.5 | 86.6 |
| | All | | | 89.5 | 88.5 | 86.5 | 86.0 | 86.7 | 85.8 | 85.3 | 83.9 |
| It | Lat | 94.2 | 93.1 | | | 89.4 | 89.1 | 89.7 | 88.9 | 90.2 | 87.4 |
| | All | 93.8 | 93.0 | | | 90.3 | 88.8 | 89.7 | 88.9 | 89.0 | 87.3 |
| Es | Lat | 93.2 | 92.0 | 94.3 | 93.3 | | | 84.7 | 84.2 | 88.5 | 88.4 |
| | All | 94.1 | 92.5 | 94.0 | 92.9 | | | 84.3 | 83.2 | 88.6 | 87.5 |
| Pt | Lat | 93.5 | 92.5 | 95.2 | 94.4 | 92.9 | 91.4 | | | 89.5 | 88.8 |
| | All | 93.7 | 92.9 | 95.2 | 93.7 | 93.0 | 90.9 | | | 88.7 | 88.6 |
| Fr | Lat | 94.4 | 93.3 | 94.7 | 94.3 | 93.8 | 91.9 | 92.7 | 92.9 | | |
| | All | 94.7 | 93.0 | 94.7 | 93.8 | 94.0 | 91.5 | 93.5 | 92.5 | | |

Table 9: Classification **accuracy** on the **test** set using the **Transformer**-based models, trained either using graphic representations (Gr), or phonetic representations (Ph). Scores above the main diagonal correspond to the Levenshtein distance-based negative samples selection, while scores below the main diagonal correspond to the random selection. Scores are averaged over 5 independent experiments using different seeds for the random engine.

| | | Ro | | It | | Es | | Pt | | Fr | |
|---|---|---|---|---|---|---|---|---|---|---|---|
| | | Gr | Ph | Gr | Ph | Gr | Ph | Gr | Ph | Gr | Ph |
| Ro | Lat | | | 89.0 | 87.2 | 87.7 | 86.6 | 85.7 | 85.1 | 87.9 | 86.1 |
| | All | | | 88.0 | 86.9 | 84.6 | 84.4 | 85.9 | 84.8 | 85.1 | 85.1 |
| It | Lat | 93.2 | 92.1 | | | 91.3 | 89.4 | 88.2 | 86.8 | 89.9 | 88.2 |
| | All | 93.9 | 91.4 | | | 89.7 | 89.1 | 87.4 | 86.6 | 89.0 | 88.7 |
| Es | Lat | 92.6 | 90.2 | 94.4 | 92.2 | | | 83.7 | 85.2 | 88.8 | 87.8 |
| | All | 93.0 | 90.7 | 91.9 | 91.5 | | | 82.5 | 83.7 | 86.4 | 85.0 |
| Pt | Lat | 93.0 | 90.6 | 93.7 | 92.9 | 91.8 | 89.4 | | | 89.1 | 88.6 |
| | All | 92.3 | 91.4 | 93.6 | 91.3 | 91.8 | 89.8 | | | 88.9 | 87.8 |
| Fr | Lat | 93.9 | 93.1 | 92.8 | 93.3 | 92.3 | 90.3 | 91.4 | 91.4 | | |
| | All | 93.4 | 91.8 | 93.9 | 91.9 | 92.2 | 88.9 | 92.5 | 90.9 | | |

Table 10: Classification **precision** on the **test** set using the **Transformer**-based models, trained either using graphic representations (Gr), or phonetic representations (Ph). Scores above the main diagonal correspond to the Levenshtein distance-based negative samples selection, while scores below the main diagonal correspond to the random selection. Scores are averaged over 5 independent experiments using different seeds for the random engine.

| | | Ro | | It | | Es | | Pt | | Fr | |
|---|---|---|---|---|---|---|---|---|---|---|---|
| | | Gr | Ph | Gr | Ph | Gr | Ph | Gr | Ph | Gr | Ph |
| Ro | Lat | | | 91.0 | 91.4 | 88.2 | 88.3 | 88.6 | 89.6 | 87.5 | 87.9 |
| | All | | | 90.9 | 89.9 | 88.9 | 88.0 | 87.3 | 86.8 | 85.4 | 82.1 |
| It | Lat | 95.0 | 94.0 | | | 87.6 | 89.3 | 92.0 | 92.0 | 90.8 | 86.8 |
| | All | 93.3 | 94.6 | | | 90.4 | 87.7 | 92.5 | 91.7 | 88.9 | 85.2 |
| Es | Lat | 94.2 | 94.5 | 94.5 | 94.9 | | | 86.6 | 83.2 | 88.9 | 90.0 |
| | All | 95.3 | 94.5 | 96.2 | 94.2 | | | 86.2 | 81.6 | 91.0 | 90.2 |
| Pt | Lat | 93.8 | 94.4 | 97.0 | 96.4 | 94.5 | 94.0 | | | 90.3 | 89.5 |
| | All | 95.2 | 94.5 | 96.9 | 96.3 | 94.0 | 91.9 | | | 88.2 | 89.6 |
| Fr | Lat | 95.3 | 93.7 | 97.1 | 95.7 | 96.0 | 94.4 | 94.6 | 94.9 | | |
| | All | 96.2 | 94.3 | 95.4 | 96.0 | 95.8 | 94.3 | 94.7 | 94.3 | | |

Table 11: Classification **recall** on the **test** set using the **Transformer**-based models, trained either using graphic representations (Gr), or phonetic representations (Ph). Scores above the main diagonal correspond to the Levenshtein distance-based negative samples selection, while scores below the main diagonal correspond to the random selection. Scores are averaged over 5 independent experiments using different seeds for the random engine.

| | | Ro | | It | | Es | | Pt | | Fr | |
|---|---|---|---|---|---|---|---|---|---|---|---|
| | | Gr | Ph | Gr | Ph | Gr | Ph | Gr | Ph | Gr | Ph |
| Ro | Lat | | | 86.9 | 86.0 | 87.2 | 83.9 | 85.3 | 83.9 | 84.1 | 82.8 |
| | All | | | 83.8 | 81.1 | 86.0 | 78.8 | 82.7 | 80.6 | 83.9 | 82.4 |
| It | Lat | 89.0 | 84.9 | | | 84.5 | 83.8 | 84.4 | 83.4 | 87.1 | 85.2 |
| | All | 86.9 | 82.3 | | | 81.8 | 81.2 | 84.0 | 82.4 | 85.5 | 83.7 |
| Es | Lat | 87.3 | 80.2 | 85.2 | 83.2 | | | 81.5 | 80.5 | 84.8 | 81.7 |
| | All | 85.6 | 76.0 | 82.3 | 81.8 | | | 79.5 | 78.1 | 83.9 | 78.9 |
| Pt | Lat | 88.9 | 83.3 | 85.2 | 82.2 | 84.4 | 79.4 | | | 86.9 | 85.0 |
| | All | 85.9 | 81.5 | 83.7 | 81.3 | 82.7 | 78.9 | | | 85.5 | 84.2 |
| Fr | Lat | 81.6 | 79.4 | 85.9 | 83.6 | 87.2 | 77.4 | 85.7 | 82.6 | | |
| | All | 82.4 | 82.0 | 83.7 | 82.3 | 86.8 | 76.7 | 83.1 | 83.0 | | |

Table 12: Classification **accuracy** on the **test** set using the **Siamese CNN** models, trained either using graphic representations (Gr), or phonetic representations (Ph). Scores above the main diagonal correspond to the Levenshtein distance-based negative samples selection, while scores below the main diagonal correspond to the random selection. Scores are averaged over 5 independent experiments using different seeds for the random engine.

|  |  | Ro | | It | | Es | | Pt | | Fr | |
|---|---|---|---|---|---|---|---|---|---|---|---|
|  |  | Gr | Ph | Gr | Ph | Gr | Ph | Gr | Ph | Gr | Ph |
| Ro | Lat |  |  | 85.6 | 84.6 | 88.9 | 82.8 | 83.0 | 82.3 | 85.4 | 83.0 |
|  | All |  |  | 83.3 | 79.4 | 84.8 | 77.2 | 81.0 | 78.1 | 85.0 | 82.8 |
| It | Lat | 86.7 | 83.6 |  |  | 82.6 | 81.3 | 85.0 | 82.6 | 87.4 | 83.9 |
|  | All | 85.4 | 80.5 |  |  | 78.2 | 77.7 | 85.0 | 83.6 | 86.1 | 83.5 |
| Es | Lat | 89.8 | 80.0 | 83.1 | 81.6 |  |  | 80.1 | 79.0 | 86.2 | 84.0 |
|  | All | 83.5 | 75.5 | 80.8 | 79.1 |  |  | 78.9 | 77.3 | 83.5 | 77.3 |
| Pt | Lat | 89.6 | 80.9 | 88.5 | 81.0 | 85.3 | 76.7 |  |  | 88.2 | 86.5 |
|  | All | 86.2 | 79.6 | 83.1 | 83.5 | 82.5 | 77.9 |  |  | 85.9 | 86.0 |
| Fr | Lat | 84.7 | 80.3 | 83.5 | 81.2 | 88.3 | 76.0 | 84.0 | 79.5 |  |  |
|  | All | 79.1 | 80.2 | 80.1 | 79.4 | 86.5 | 74.3 | 81.2 | 81.7 |  |  |

Table 13: Classification **precision** on the **test** set using the **Siamese CNN** models, trained either using graphic representations (Gr), or phonetic representations (Ph). Scores above the main diagonal correspond to the Levenshtein distance-based negative samples selection, while scores below the main diagonal correspond to the random selection. Scores are averaged over independent experiments using different seeds for the random engine.

|  |  | Ro | | It | | Es | | Pt | | Fr | |
|---|---|---|---|---|---|---|---|---|---|---|---|
|  |  | Gr | Ph | Gr | Ph | Gr | Ph | Gr | Ph | Gr | Ph |
| Ro | Lat |  |  | 88.5 | 87.7 | 85.4 | 86.7 | 88.1 | 85.3 | 83.3 | 83.8 |
|  | All |  |  | 84.1 | 82.9 | 87.3 | 81.7 | 85.3 | 84.7 | 82.2 | 81.8 |
| It | Lat | 91.6 | 86.1 |  |  | 88.2 | 88.7 | 84.6 | 85.5 | 87.1 | 88.0 |
|  | All | 88.2 | 84.4 |  |  | 87.1 | 86.4 | 82.1 | 80.3 | 84.5 | 83.9 |
| Es | Lat | 84.6 | 81.3 | 89.0 | 86.5 |  |  | 84.7 | 84.1 | 83.9 | 80.2 |
|  | All | 88.3 | 76.2 | 83.4 | 85.1 |  |  | 79.9 | 78.3 | 83.5 | 80.2 |
| Pt | Lat | 87.3 | 86.1 | 81.4 | 84.6 | 83.6 | 85.0 |  |  | 86.0 | 83.5 |
|  | All | 85.0 | 84.4 | 84.0 | 77.3 | 82.2 | 79.9 |  |  | 85.0 | 82.0 |
| Fr | Lat | 77.9 | 78.8 | 90.0 | 88.1 | 86.6 | 82.0 | 88.6 | 88.7 |  |  |
|  | All | 87.7 | 85.6 | 89.1 | 86.7 | 86.4 | 79.8 | 86.0 | 85.0 |  |  |

Table 14: Classification **recall** on the **test** set using the **Siamese CNN** models, trained either using graphic representations (Gr), or phonetic representations (Ph). Scores above the main diagonal correspond to the Levenshtein distance-based negative samples selection, while scores below the main diagonal correspond to the random selection. Scores are averaged over independent experiments using different seeds for the random engine.

|  |  | Ro | | | It | | | Es | | | Pt | | | Fr | | |
|---|---|---|---|---|---|---|---|---|---|---|---|---|---|---|---|---|
|  |  | Gr | Ph | En | Gr | Ph | En | Gr | Ph | En | Gr | Ph | En | Gr | Ph | En |
| Ro | Lat |  |  |  | 93.0 | 93.2 | 94.0 | 90.1 | 89.9 | 91.0 | 90.4 | 90.5 | 92.3 | 90.2 | 89.1 | 91.3 |
|  | All |  |  |  | 92.5 | 92.6 | 94.3 | 90.3 | 88.6 | 90.3 | 89.7 | 89.7 | 91.2 | 89.6 | 87.8 | 90.6 |
| It | Lat | 98.6 | 97.9 | 98.6 |  |  |  | 92.9 | 92.6 | 94.1 | 93.4 | 93.3 | 94.7 | 91.5 | 92.2 | 94.0 |
|  | All | 98.6 | 98.1 | 98.7 |  |  |  | 93.2 | 93.1 | 94.0 | 93.8 | 93.4 | 95.0 | 92.4 | 92.1 | 93.9 |
| Es | Lat | 98.2 | 97.4 | 98.2 | 98.2 | 98.1 | 98.6 |  |  |  | 90.4 | 89.6 | 91.6 | 92.2 | 91.8 | 93.9 |
|  | All | 97.9 | 97.4 | 98.3 | 98.3 | 97.8 | 98.7 |  |  |  | 91.0 | 88.8 | 91.3 | 92.4 | 91.8 | 93.9 |
| Pt | Lat | 97.8 | 96.9 | 98.3 | 98.6 | 98.0 | 98.8 | 97.6 | 97.0 | 98.2 |  |  |  | 92.0 | 92.4 | 94.2 |
|  | All | 98.1 | 97.8 | 98.6 | 98.5 | 98.3 | 98.9 | 97.6 | 97.2 | 98.3 |  |  |  | 92.1 | 91.8 | 93.5 |
| Fr | Lat | 98.2 | 97.7 | 98.5 | 98.5 | 98.1 | 98.8 | 97.6 | 97.3 | 98.4 | 97.7 | 97.7 | 98.4 |  |  |  |
|  | All | 98.2 | 97.8 | 98.7 | 98.5 | 98.3 | 98.9 | 97.9 | 97.5 | 98.5 | 97.8 | 97.9 | 98.4 |  |  |  |

Table 15: Mean classification **accuracy** of the **3-fold cross validation** experiment for **ensemble** models, trained either exclusively using graphic classifiers (Gr), or phonetic classifiers (Ph) , or a combination of both (En). Scores above the main diagonal correspond to the Levenshtein distance-based negative samples selection, while scores below the main diagonal correspond to the random selection.

| | | Ro | | | It | | | Es | | | Pt | | | Fr | | |
|---|---|---|---|---|---|---|---|---|---|---|---|---|---|---|---|---|
| | | Gr | Ph | En | Gr | Ph | En | Gr | Ph | En | Gr | Ph | En | Gr | Ph | En |
| Ro | Lat | | | | 92.9 | 93.8 | 95.2 | 90.4 | 89.6 | 92.0 | 88.8 | 91.3 | 92.5 | 90.4 | 89.8 | 91.6 |
| | All | | | | 92.5 | 92.9 | 94.3 | 89.0 | 88.4 | 90.6 | 89.8 | 90.2 | 91.6 | 87.8 | 88.6 | 90.8 |
| It | Lat | 98.7 | 97.9 | 98.7 | | | | 93.3 | 93.0 | 94.5 | 94.2 | 93.8 | 95.0 | 93.2 | 93.4 | 94.8 |
| | All | 98.7 | 98.3 | 98.8 | | | | 93.0 | 93.0 | 93.8 | 94.1 | 93.7 | 95.2 | 92.4 | 93.0 | 94.3 |
| Es | Lat | 98.3 | 97.6 | 98.2 | 98.3 | 98.1 | 98.7 | | | | 91.0 | 89.9 | 92.0 | 92.4 | 93.0 | 94.4 |
| | All | 98.1 | 97.9 | 98.5 | 98.3 | 97.8 | 98.7 | | | | 89.7 | 88.8 | 91.1 | 91.5 | 92.8 | 94.2 |
| Pt | Lat | 98.1 | 97.1 | 98.3 | 98.7 | 98.2 | 98.9 | 97.7 | 97.2 | 98.4 | | | | 92.6 | 93.6 | 95.0 |
| | All | 98.1 | 98.0 | 98.7 | 98.6 | 98.4 | 99.0 | 97.7 | 97.5 | 98.4 | | | | 91.7 | 93.3 | 94.4 |
| Fr | Lat | 98.4 | 98.1 | 98.7 | 98.6 | 98.4 | 99.0 | 97.8 | 97.6 | 98.7 | 97.7 | 97.8 | 98.7 | | | |
| | All | 98.3 | 98.1 | 98.8 | 98.8 | 98.5 | 99.1 | 98.1 | 97.8 | 98.7 | 97.9 | 98.2 | 98.6 | | | |

Table 16: Mean classification **precision** of the **3-fold cross validation** experiment for **ensemble** models, trained either exclusively using graphic classifiers (Gr), or phonetic classifiers (Ph) , or a combination of both (En). Scores above the main diagonal correspond to the Levenshtein distance-based negative samples selection, while scores below the main diagonal correspond to the random selection.

| | | Ro | | | It | | | Es | | | Pt | | | Fr | | |
|---|---|---|---|---|---|---|---|---|---|---|---|---|---|---|---|---|
| | | Gr | Ph | En | Gr | Ph | En | Gr | Ph | En | Gr | Ph | En | Gr | Ph | En |
| Ro | Lat | | | | 93.1 | 92.7 | 92.7 | 89.6 | 90.2 | 89.8 | 92.7 | 89.7 | 92.3 | 89.9 | 88.1 | 90.9 |
| | All | | | | 92.5 | 92.4 | 94.3 | 92.0 | 89.0 | 90.0 | 89.5 | 89.1 | 90.8 | 92.1 | 86.9 | 90.4 |
| It | Lat | 98.5 | 97.9 | 98.5 | | | | 92.3 | 92.0 | 93.6 | 92.4 | 92.7 | 94.3 | 89.5 | 90.8 | 93.2 |
| | All | 98.6 | 97.9 | 98.5 | | | | 93.5 | 93.3 | 94.4 | 93.5 | 93.0 | 94.7 | 92.5 | 91.0 | 93.5 |
| Es | Lat | 98.0 | 97.1 | 98.1 | 98.0 | 98.0 | 98.5 | | | | 89.5 | 89.1 | 91.2 | 92.0 | 90.3 | 93.2 |
| | All | 97.7 | 97.0 | 98.0 | 98.3 | 97.8 | 98.7 | | | | 92.6 | 88.9 | 91.7 | 93.6 | 90.8 | 93.5 |
| Pt | Lat | 97.5 | 96.8 | 98.2 | 98.4 | 97.9 | 98.7 | 97.4 | 96.8 | 98.1 | | | | 91.3 | 91.0 | 93.2 |
| | All | 98.1 | 97.5 | 98.6 | 98.5 | 98.1 | 98.8 | 97.5 | 97.0 | 98.2 | | | | 92.5 | 90.0 | 92.4 |
| Fr | Lat | 97.9 | 97.2 | 98.2 | 98.4 | 97.9 | 98.6 | 97.3 | 96.8 | 98.1 | 97.6 | 97.5 | 98.2 | | | |
| | All | 98.1 | 97.4 | 98.5 | 98.3 | 98.2 | 98.7 | 97.7 | 97.2 | 98.3 | 97.7 | 97.6 | 98.3 | | | |

Table 17: Mean classification **recall** of the **3-fold cross validation** experiment for **ensemble** models, trained either exclusively using graphic classifiers (Gr), or phonetic classifiers (Ph) , or a combination of both (En). Scores above the main diagonal correspond to the Levenshtein distance-based negative samples selection, while scores below the main diagonal correspond to the random selection.

| | | Ro | | It | | Es | | Pt | | Fr | |
|---|---|---|---|---|---|---|---|---|---|---|---|
| | | Gr | Ph | Gr | Ph | Gr | Ph | Gr | Ph | Gr | Ph |
| Ro | Lat | | | 89.9 | 88.9 | 86.5 | 86.2 | 87.6 | 86.9 | 87.3 | 85.4 |
| | All | | | 89.6 | 89.2 | 87.6 | 85.6 | 86.2 | 86.1 | 84.7 | 83.1 |
| It | Lat | 94.9 | 93.6 | | | 89.7 | 89.8 | 89.6 | 89.3 | 90.3 | 88.8 |
| | All | 94.6 | 93.5 | | | 90.2 | 89.2 | 90.2 | 89.1 | 88.3 | 86.7 |
| Es | Lat | 93.7 | 91.4 | 94.1 | 93.0 | | | 86.3 | 85.1 | 89.0 | 88.9 |
| | All | 94.8 | 93.1 | 94.1 | 93.0 | | | 85.4 | 85.1 | 88.3 | 86.8 |
| Pt | Lat | 92.2 | 91.1 | 94.4 | 93.8 | 92.8 | 91.8 | | | 89.5 | 87.0 |
| | All | 93.9 | 92.5 | 94.9 | 93.5 | 92.4 | 90.5 | | | 88.3 | 87.3 |
| Fr | Lat | 94.2 | 93.3 | 94.7 | 93.7 | 93.9 | 92.8 | 93.1 | 93.1 | | |
| | All | 95.2 | 93.7 | 95.3 | 93.9 | 94.1 | 92.5 | 93.8 | 92.8 | | |

Table 18: Classification **accuracy** on the **validation** set using the **Transformer**-based models, trained either using graphic representations (Gr), or phonetic representations (Ph). Scores above the main diagonal correspond to the Levenshtein distance-based negative samples selection, while scores below the main diagonal correspond to the random selection. Scores are averaged over 5 independent experiments using different seeds for the random engine.

|  |  | Ro | | It | | Es | | Pt | | Fr | |
|---|---|---|---|---|---|---|---|---|---|---|---|
|  |  | Gr | Ph | Gr | Ph | Gr | Ph | Gr | Ph | Gr | Ph |
| Ro | Lat |  |  | 89.9 | 87.9 | 86.0 | 85.2 | 87.4 | 86.7 | 87.7 | 84.7 |
|  | All |  |  | 87.8 | 87.5 | 86.3 | 85.1 | 85.5 | 85.0 | 82.8 | 82.7 |
| It | Lat | 94.8 | 93.2 |  |  | 90.2 | 89.5 | 87.1 | 86.5 | 88.9 | 88.7 |
|  | All | 93.7 | 91.0 |  |  | 90.2 | 89.6 | 88.0 | 86.8 | 86.9 | 86.7 |
| Es | Lat | 92.2 | 89.7 | 93.3 | 91.0 |  |  | 84.9 | 85.9 | 88.3 | 87.2 |
|  | All | 93.4 | 91.3 | 93.1 | 91.9 |  |  | 84.7 | 86.5 | 86.9 | 85.6 |
| Pt | Lat | 92.2 | 90.0 | 92.5 | 92.2 | 92.1 | 90.2 |  |  | 88.4 | 85.5 |
|  | All | 92.1 | 90.4 | 93.7 | 91.4 | 91.6 | 89.7 |  |  | 87.5 | 84.8 |
| Fr | Lat | 92.4 | 92.3 | 93.0 | 91.5 | 92.5 | 90.9 | 91.6 | 91.7 |  |  |
|  | All | 93.5 | 92.7 | 94.4 | 92.1 | 93.2 | 91.5 | 92.5 | 90.7 |  |  |

Table 19: Classification **precision** on the **validation** set using the **Transformer**-based models, trained either using graphic representations (Gr), or phonetic representations (Ph). Scores above the main diagonal correspond to the Levenshtein distance-based negative samples selection, while scores below the main diagonal correspond to the random selection. Scores are averaged over 5 independent experiments using different seeds for the random engine.

|  |  | Ro | | It | | Es | | Pt | | Fr | |
|---|---|---|---|---|---|---|---|---|---|---|---|
|  |  | Gr | Ph | Gr | Ph | Gr | Ph | Gr | Ph | Gr | Ph |
| Ro | Lat |  |  | 90.6 | 91.0 | 87.1 | 87.6 | 88.2 | 87.6 | 86.5 | 86.0 |
|  | All |  |  | 91.2 | 90.6 | 88.9 | 85.9 | 86.0 | 86.5 | 86.4 | 82.4 |
| It | Lat | 95.4 | 94.6 |  |  | 88.8 | 89.8 | 92.8 | 93.0 | 91.2 | 88.0 |
|  | All | 95.4 | 96.2 |  |  | 90.4 | 88.9 | 93.2 | 92.3 | 89.1 | 85.4 |
| Es | Lat | 95.5 | 93.5 | 94.9 | 95.3 |  |  | 87.9 | 83.6 | 90.2 | 91.5 |
|  | All | 96.2 | 95.2 | 95.4 | 94.5 |  |  | 87.3 | 83.9 | 90.7 | 89.0 |
| Pt | Lat | 92.5 | 92.6 | 96.6 | 95.7 | 93.6 | 93.5 |  |  | 91.0 | 89.0 |
|  | All | 95.6 | 94.5 | 96.3 | 96.0 | 93.6 | 92.0 |  |  | 88.3 | 89.7 |
| Fr | Lat | 96.2 | 94.5 | 96.3 | 95.8 | 95.7 | 95.3 | 94.8 | 94.8 |  |  |
|  | All | 97.0 | 94.6 | 96.0 | 95.4 | 95.4 | 94.0 | 94.8 | 94.8 |  |  |

Table 20: Classification **recall** on the **validation** set using the **Transformer**-based models, trained either using graphic representations (Gr), or phonetic representations (Ph). Scores above the main diagonal correspond to the Levenshtein distance-based negative samples selection, while scores below the main diagonal correspond to the random selection. Scores are averaged over 5 independent experiments using different seeds for the random engine.