# OpenReview forum: "RoBoCoP: A Comprehensive ROmance BOrrowing COgnate Package and Benchmark for Multilingual Cognate Identification"
_EMNLP/2023/Conference — EMNLP 2023 Main_

### Official Review · Reviewer_kDVS · 2023-08-05

**Soundness:** 4

**Excitement:**

4: Strong: This paper deepens the understanding of some phenomenon or lowers the barriers to an existing research direction.

**Paper Topic And Main Contributions:**

The paper introduces a comprehensive database of Romance cognate pairs and borrowings based on the etymological information provided by the dictionaries. The pairs of cognates were extracted between any two Romance languages by parsing electronic dictionaries of Romanian, Italian, Spanish, Portuguese and French.
The paper also provides the benchmark for automatic cognate detection.

**Questions For The Authors:**

How do you define the cognate pairs and Borrowings? Are you considering the historical derivatives in borrowing as well?

**Reasons To Accept:**

The paper is well-documented and written.
The paper introduces a new database for cognate and borrowings, which is a notable contribution to developing a corpus. This database could be further extended and help study the phenomena of cognate detection and many more NLP tasks.
The experiments were executed nicely, and the findings were explained.


**Reasons To Reject:**

I could not find any reason to reject the paper.


**Reproducibility:**

4: Could mostly reproduce the results, but there may be some variation because of sample variance or minor variations in their interpretation of the protocol or method.

**Reviewer Confidence:**

4: Quite sure. I tried to check the important points carefully. It's unlikely, though conceivable, that I missed something that should affect my ratings.

---

> ### Author Rebuttal · Authors · 2023-08-29
>
> Thank you for your review and positive comments on our work.
>
> Our working definition for cognate pairs is the one formulated on lines 312-318 in the paper.
> In this phase of the project, we have considered all the derivatives as borrowings as well, for the mere reason that it needed to be an exhaustive lexical database. These derivatives appear as internal creations, i.e., words that, regardless of the source language they were borrowed from, have given birth to new words, thus derivatives from a lexeme that was already integrated in the target language.

---

### Official Review · Reviewer_U5uj · 2023-08-05

**Soundness:** 3

**Excitement:**

3: Ambivalent: It has merits (e.g., it reports state-of-the-art results, the idea is nice), but there are key weaknesses (e.g., it describes incremental work), and it can significantly benefit from another round of revision. However, I won't object to accepting it if my co-reviewers champion it.

**Missing References:**

Brown, Cecil H., Holman, Eric W., Wichmann, Søren and Velupillai, Viveka. "Automated classification of the world′s languages: a description of the method and preliminary results" Language Typology and Universals, vol. 61, no. 4, 2008, pp. 285-308. https://doi.org/10.1524/stuf.2008.0026

The discussions on AJSP might be worth looking at as they devise a particular orthography for representing word lists.

**Paper Topic And Main Contributions:**

This paper contributes a large database of cognate pairs and borrowed words for 5 Romance languages. It is much larger than any of the previous cognate databases for these languages. As per the authors, the goal of this work is to make a contribution to both historical linguistics studies as well as computational linguistics.

**Questions For The Authors:**


A. The examples provided in lines 131 and 132 are unclear: What is the ambiguity or issue being discussed here with respect to etymology ?
B. What is the accuracy of the eSpeak library in grapheme to phoneme conversion for these languages, especially as these are isolated words?
C. Was the graphemic transcription of the words in the database Latin alphabet without diacritics ? What exactly was the form of transcription for all the 5 languages?
C. In Table 1, the Spanish column also shows a source language that is Spanish: this is probably an error/typo?

**Reasons To Accept:**

The paper describes a large and laborious task of extracting cognate + borrowed word information from existing dictionaries. It results in a large database of cognate + borrowed word pairs for 5 closely related languages. The authors also test a variety of machine learning models, and show high accuracy for the task. The experimental setup has been done carefully and review of previous work is quite thorough.

**Reasons To Reject:**

The authors show that the database includes both cognates and borrowed word pairs- but in lines 86-94, they discuss a 'broadened' definition of cognates. If the primary motivation to 'obtain longer lists of cognate sets' (line 86) - then does it serve the goals of computational lingustics or historical linguistics -- or both? I believe the author is trying to show that it can serve both, but while pre-processing the data, all special characters indicating pronunciation are removed (line 515,253). Phonological information would be important for the task of comparative analysis. If the grapheme to phoneme representation/conversion has very high accuracy then perhaps there's no problem but this has not been discussed in the paper.

Further, it would be interesting to see how these models perform on cognate pairs that are very lexically divergent especially function words e.g. wh-words like 'What' or 'when'. How do these models work with this type of data? Particularly it might be interesting to compare the transformer models with the ensemble models with these cases.
Given the amount of previous work done on cognate detection it is not surprising to see good results for this 'similarity/distance' based task. It would have been nice if the authors could have given a little more space to the pipeline described in Section 2 and 2.1 as it would reveal some crucial decisions taken while building this database from the dictionaries.


**Reproducibility:**

4: Could mostly reproduce the results, but there may be some variation because of sample variance or minor variations in their interpretation of the protocol or method.

**Reviewer Confidence:**

5: Positive that my evaluation is correct. I read the paper very carefully and I am very familiar with related work.

**Typos Grammar Style And Presentation Improvements:**

Line 74 (counter economic ?)
Citation style in section 3.1 (in line citation not followed)
Better indentation throughout the paper would be helpful

---

> ### Author Rebuttal · Authors · 2023-08-29
>
> We thank you for your review and your suggestions.
>
> A. Here we discuss the phonetic descendance of Latin initial /s-/, normally transmitted as an /s-/ as well in the Romance languages, but which sometimes has irregular results, such as /ch-/ [tꭍ] in Spanish. For this reason, the etymology of chillar 'to squeak' still appears in Diccionario de la lengua española (DRAE) as derived from Lat. fistulare 'to play the flute', although [1] demonstrated that it comes from siflare 'to whistle'.
>
> B. The espeak library’s implementation relies both on dictionary-based and rule-based approaches. Thus the transcription dictionaries contain in most cases the exact transcription of the words in our database, which are written in their corresponding language’s dictionary form.
>
> C. When converting the graphic representation of the words to their corresponding phonetic representation, the accents/diacritics were not stripped, in order to obtain a reliable phonetic output. Our database records the words written in Latin alphabet with diacritics and accents (where they are part of the word's spelling, and not special symbols). Accent removal was applied only in the feature extraction algorithm, when computing alignment patterns for the graphic representations. Furthermore, we preserve in the database as a separate column the raw form of the word/etymon as it was found in the dictionary (before normalization).
>
> D. Indeed, the etymon dictionaries for each language may include words formed internally in the language (e.g. derived words) - for these the language of the etymon is the same as the language of the word, as we must consider the basic word as already part of the target language.
>
> With regards to your comment on the specific details of data cleaning when parsing the dictionaries: the process was very specific to each dictionary and included a cyclical process similar to methodologies used in web scraping - running scripts implementing rule-based algorithms (such as regular expressions) to separate noise from the data for each dictionary and manual evaluation of each output with the assistance of linguists in our team, followed by potential refinement of the code to manage all exceptions. We will include some examples in the Appendix of the final version for a better illustration of the kinds of problems encountered.
>
>
> [1] Malkiel, Yakov, Diachronic problems in Pholosymbolism. Edita and inedita, 1979-1988, vol. I, Amsterdam/Philadelphia, John Benjamins.

---

### Official Review · Reviewer_HFjy · 2023-08-05

**Typos Grammar Style And Presentation Improvements:** The paper is generally well-structure…
**Soundness:** 4

**Excitement:**

4: Strong: This paper deepens the understanding of some phenomenon or lowers the barriers to an existing research direction.

**Missing References:**

Mitkov, R., Pekar, V., Blagoev, D., & Mulloni, A. (2007). Methods for Extracting and Classifying Pairs of Cognates and False Friends. Machine Translation, 21(1), 29–53. http://www.jstor.org/stable/30219109



**Paper Topic And Main Contributions:**

The paper presents a benchmark for the automatic detection of cognates from electronic dictionaries in Romance languages using ML techniques. The dataset comprises a database of (pairs of) Romance cognates and borrowings based on the etymological information included in electronic dictionaries in French, Italian, Portuguese, Romanian, and Spanish. The pairs of cognates were extracted using machine learning techniques and deep learning methods. This research is of interest to historical linguistics and digital humanities, whereas, the results reported are promising.
The main contributions of the paper are the following:
- a database of cognates for five Romance languages;
- a method for extracting the cognates from dictionaries.




**Questions For The Authors:**

l. 220 a fully available: publicly available?

**Reasons To Accept:**

The main strengths of the paper can be summarised as follows:
- a complete presentation of the notion of cognates;
- a resource that could possibly be of use to future research in historical linguistics;
- the human-on-the-loop process ensures the quality of the data;
- the authors provide meaningful comparisons with similar databases;
- the methodology employed and the experimental setting are presented clearly.

**Reasons To Reject:**

There are no serious weaknesses noticed. However, reference to the methodology used for parsing the dictionary and extracting the words and their etymons as well as the efficacy of the process would be of interest.

**Reproducibility:**

4: Could mostly reproduce the results, but there may be some variation because of sample variance or minor variations in their interpretation of the protocol or method.

**Reviewer Confidence:**

4: Quite sure. I tried to check the important points carefully. It's unlikely, though conceivable, that I missed something that should affect my ratings.

---

> ### Author Rebuttal · Authors · 2023-08-29
>
> Thank you very much for your review and positive evaluation of our work.
>
> We intend to make the full contents of the package available for research purposes upon request (including word and etymon lists for each language, with graphic and phonetic forms, as well as cognate pairs for each language pair). Thank you for the question - we will clarify this in the final version. We will amend the final version and we will cite the missing reference as well.
>
> With regards to your comment on the specific details of data cleaning when parsing the dictionaries: the process was very specific to each dictionary and included a cyclical process similar to methodologies used in web scraping - running scripts implementing rule-based algorithms (such as regular expressions) to separate noise from the data for each dictionary and manual evaluation of each output with the assistance of linguists in our team, followed by potential refinement of the code to manage all exceptions. We will include some examples in the Appendix of the final version for a better illustration of the kinds of problems encountered.

---

### Meta-Review · Area_Chair_zcL6 · 2023-09-09

**Recommendation:** 5

**Metareview:**

This paper introduces a corpus of cognate pairs and borrowed words for 5 Romance languages, the largest known corpus of its kind, with potential impact for the field of historical and computational linguistics. The content and presentation of this paper are of good quality, as reflected by its Soundness (three 4s, one 3), Excitement (two 4s, one 3), and Reproducibility scores (two 4s, one 5). In particular, the paper was praised for its literature review, the quality of the corpus presented (One reviewer commended the “human-on-the-loop process”), and sound experimental results that present a solid benchmark for the automatic detection of cognates. Only one reviewer cited reasons to reject the paper, questioning some aspects of the data collection methodology (e.g. the removal of special characters that indicate pronunciation). These concerns, alongside general comments and questions from all reviewers, were resolved during the author rebuttal period to reviewer satisfaction.
In light of this, only minor revisions, addressing reviewers’ comments and questions, need to be made to ensure this paper is camera ready.

---

### Decision · Program_Chairs · 2023-10-07

**Decision:**

Accept-Main

**Comment:**

This paper introduces a corpus of cognate pairs and borrowed words for 5 Romance languages, the largest known corpus of its kind, with potential impact for the field of historical and computational linguistics. The content and presentation of this paper are of good quality, as reflected by its Soundness (three 4s, one 3), Excitement (two 4s, one 3), and Reproducibility scores (two 4s, one 5). In particular, the paper was praised for its literature review, the quality of the corpus presented (One reviewer commended the “human-on-the-loop process”), and sound experimental results that present a solid benchmark for the automatic detection of cognates. Only one reviewer cited reasons to reject the paper, questioning some aspects of the data collection methodology (e.g. the removal of special characters that indicate pronunciation). These concerns, alongside general comments and questions from all reviewers, were resolved during the author rebuttal period to reviewer satisfaction.
In light of this, only minor revisions, addressing reviewers’ comments and questions, need to be made to ensure this paper is camera ready.